# The Effect of Various Salinities and Light Intensities on the Growth Performance of Five Locally Isolated Microalgae [*Amphidinium carterae*, *Nephroselmis* sp., *Tetraselmis* sp. (var. red pappas), *Asteromonas gracilis* and *Dunaliella* sp.] in Laboratory Batch Cultures

**George N. Hotos** *[ID] **and Despoina Avramidou**





Plankton Culture Laboratory, Department of Animal Production, Fisheries and Aquaculture, University of Patras, 30200 Messolonghi, Greece; dabramid@upatras.gr
* Correspondence: ghotos@upatras.gr

**Abstract:** After a 1.5-year screening survey in the lagoons of Western Greece in order to isolate and culture sturdy species of microalgae for aquaculture or other value-added uses, as dictated primarily by satisfactory potential for their mass culture, five species emerged, and their growth was monitored in laboratory conditions. *Amphidinium carterae*, *Nephroselmis* sp., *Tetraselmis* sp. (var. red pappas), *Asteromonas gracilis*, and *Dunaliella* sp. were batch cultured using low (20 ppt), sea (40 ppt), and high salinity (50 or 60 or 100 ppt) and in combination with low (2000 lux) and high (8000 lux) intensity illumination. The results exhibited that all these species can be grown adequately in all salinities and with the best growth in terms of maximum cell density, specific growth rate (SGR), and biomass yield (g dry weight/L) at high illumination (8000 lux). The five species examined exhibited different responses in the salinities used, whereby *Amphidinium* clearly performs best in 20 ppt, far better than 40 ppt, and even more so than 50 ppt. *Nephroselmis* and *Tetraselmis* grow almost the same in 20 and 40 ppt and less well in 60 ppt. *Asteromonas* performs best in 100 ppt, although it can grow quite well in both 40 and 60 ppt. *Dunaliella* grows equally well in all salinities (20, 40, 60 ppt). Concerning the productivity, assessed as the maximum biomass yield at the end of the culture period, the first rank is occupied by *Nephroselmis* with ~3.0 g d.w./L, followed by *Tetraselmis* (2.0 g/L), *Dunaliella* (1.58 g/L), *Amphidinium* (1.19 g/L), and *Asteromonas* (0.7 g/L) with all values recorded at high light (8000 lux).

**Keywords:** salinity; light; growth; microalgae; *Amphidinium*; *Nephroselmis*; *Tetraselmis*; *Asteromonas*; *Dunaliella*

## 1. Introduction

As microalgae have established a foothold in our civilization in terms of an effective source of products for various uses, the quest for novel species suitable for culture or the improvement of existing culture techniques are both welcomed. In fact, nowadays we are at the start of the adventure of delving in algal culture, and new challenges emerge as to how we can extract the most benefits microalgae can offer. The modern-days problems of tackling increasing fossil fuel emissions, struggling for fresh water in an expanding human population, finding cheap energy alternatives to fossil fuels sources, creating sea farms as a relief to crop land, and replacing meat with plant-like equivalent organisms, all lead to algae and their culture. Putting aside for the moment the case of macroalgae whose semi-natural sea culture-based harvest in East Asia comprises about one-third of the global aquaculture output, we focus on microalgae as having the potential to be cultured everywhere and in numerous methods of cultivation [1].

In order to focus on target species that offer ease of culture and valuable products, the first is of paramount importance. In fact, what is the benefit if the tested species is cumber-

some in its biomass production? Moreover, the effectiveness of its culture must be seen as a prerequisite for all aspects of its exploitation. Proteins, lipids, pigments, antioxidants, and other special minor constituents of its cell, to a larger or smaller degree part of its biochemical profile, can offer materials usable in several industry sectors. So, notwithstanding the already-existing large catalogue of cultured species of microalgae, e.g., [1,2], much remains to be improved, both in the existing methods and (most importantly) the examination of new candidate species that have not yet been trial-cultured.

The first step in the culture procedure is the small-scale laboratory culture under the most basic conditions in which their manipulation is easily accomplished. Salinity and light are the primary factors for marine species and can be altered and maintained at desirable levels without dramatically upsetting the environment of either the culture medium or its exterior. Temperature is, of course, of paramount importance and it is well known that higher temperatures of 25–30 °C can boost the growth rate [3,4] but such high temperatures are costly to maintain in temperate regions and only temperatures in the range of 18–20 °C are economically feasible indoors during the cold months. So, we consider the temperature of around 20 °C as a default value in testing all other parameters, and if the experiments turn out to be promising, there is no reason to doubt that the species under experimentation can perform better in higher temperatures during the warm months. The manipulation of pH has little to offer because its control is extremely cumbersome and practically of no decisive importance, as the culture will create its own pH. Nutrients are of paramount importance for algal growth, but once a widely accepted recipe is selected that covers all the needs of microalgae, it can also be considered as a default parameter.

Light is the most essential and critical factor because directly affects the photosynthesis from which biosynthesis of biomass ensues [5]. Low lighting has a limiting effect on the growth of microalgae, so increasing the light intensity in the mass cultivation of algae is a common practice to enhance growth [6], but care should be taken as too much can cause photoinhibition [7,8]. LED light is preferable for use because it does not create extra heat that other light sources (e.g., fluorescent) do, disturbing in that way the desirable temperature regime for the experimentation (personal observation).

Salinity affects the growth of microalgae, acting directly on the osmoregulatory mechanism of the cell. All marine species of microalgae can endure several ranges of salinity but the existing information in the literature is rather complicated on conclusions about both the range of tolerance and the optimum value. There are studies suggesting that elevated salinities negatively affect the growth of microalgae, acting directly on their photosynthetic apparatus [9,10].

Only after reaching conclusions about the ensuing growth under various salinities and light intensities it is logical to proceed to more elaborate experimentation on the influence of temperature, nutrient composition, pH, the addition of extra $CO_2$, the level of aeration, etc., at the optimum salinity and light, and beyond that, the influence of these on the biochemical composition.

In the present study, four chlorophytes (*Nephroselmis* sp., *Asteromonas gracilis*, *Tetraselmis* sp. (var. red pappas), and *Dunaliella* sp.) and one dinoflagellate (*Amphidinium carterae*) were selected for culture from a wealth of species isolated from the lagoons of Western Greece after a screening program of 1.5 years during 2019–2020. The above species, after numerous renovations of their culture medium, exhibited a remarkable ability to adapt easily to indoor conditions and dominate the culture. This is a sign of feasibility for mass culturing, as in large-scale cultures, especially in open ponds, it is very difficult to attain a pure monoculture of the desired species unless the species is able to dominate in terms of growth over other undesired species (that finally find their way to intrude). As lagoons are among the harshest water environments due to intense seasonal fluctuations of salinity, light, temperature, nutrient input, and pollution agents, it is logical to assume that species encountered there are sturdy because of their adaptability to such constantly changing water bodies. Additionally, locally isolated and exploited species of microalgae are best suited for responding well to environmental conditions prevailing in the same region when

their culture is attempted. The estuarine or lagoonal species can differ in adaptability from their pelagic counterparts [11], and in terms of locality, if proved to be easily cultured, can free the local producers from the usage of imported species [12]. Furthermore, these local strains can enrich the collections of algae preserved worldwide in order to create an expanding deposit of strains of sibling or novel species.

From the five species of the present microalgae (Supplementary Material in Videos S1–S5), three of them (*Asteromonas gracilis*, *Nephroselmis* sp., and *Amphidinium carterae*) have not been studied in a concise manner in laboratory cultures and only fragmented information can be detected on them. So, our study aspires to pave the way for future attempts on more elaborate approaches. On the other hand, *Dunaliella* sp. has been extensively studied in every possible way and our study aims to add information for this local strain. The case of *Tetraselmis* sp. (var. red pappas) is quite peculiar as, on the one hand, it belongs to the family of *Tetraselmis*, a well-studied species, but on the other, its unique feature of coloring the culture medium red suggests it is a novel strain of *Tetraselmis* with its own potential and worth study.

## 2. Materials and Methods

### 2.1. Isolation and Purification

After many monthly collections of water samples from various lagoons of W. Greece, such as Messolonghi lagoon and its saltworks (prefecture of Etoloakarnania), the lagoon of Kalogria and Pappas (pref. of Achaia), and the lagoon of Kotyhi (pref. of Ilia), the samples were transported to the laboratory and 200 mL of each sample were put in glass 1-L conical Erlenmeyer flasks containing 800 mL of sterilized 40 ppt water enriched with Walne's nutrient formula [13]. The so-called maintenance flasks were left for one week to mature, supplied through a 1-mL pipette with filtered air (~0.5 flask volume/min) fed by a central blower, exposed to continuous light of 3000 lux emitted by white light LED tubes, in an air-conditioned room with a temperature of 20–22 °C. The maintenance flasks were left for 2–3 weeks to develop a microalgae population, evident by the coloration of the water and microscopic examination. Those with no sign of coloration after 3 weeks were discarded. By this practice, we deliberately focused on algae species that can either solely or in companion with other species fully adapt to seawater salinity and dominate the culture. After the confirmation of the establishment of one or more species in the maintenance flasks, serial dilutions were performed in successive steps using 20-mL glass Erlenmeyer flasks filled with sterilized and fertilized water, as above, of the same salinity (40 ppt). The inoculated flasks were left to mature for 20 days in a special thermo-regulated chamber at 19 °C in low, ambient continuous illumination of 1300 lux and were mildly hand agitated daily. After 20 days they were examined microscopically and if a monoculture was observed, the content of this flask was transferred to 500-mL flasks prepared with a new fertilized medium and left to mature (20–22 °C, continuous illumination of 3000 lux) until used for inoculation of various bigger flasks.

### 2.2. Experimental Conditions

All microalgae were batch-cultured using either 2-L conical Erlenmeyer flasks or 1-L cylindrical plastic bottles in 2 or 3 replicates for each treatment. Saline water of 20, 40 and 50 or 60 or 100 ppt, and two light intensities of 2000 and 8000 lux from 20-watt 1600 lm LED lamps, measured at the middle of the outer surface of the vessel (Lux meter BIOBLOCK LX-101), were combined so as to create 6 treatments (3 salinities × 2 light intensities). In the case of *A. carterae*, the salinities were 20, 40 and 50 ppt and for *A. gracilis* they were 40, 60 and 100 ppt because the former species cannot tolerate too-high salinities and the latter too low, as recorded in preliminary trials. The photoperiod of 16 hL:8 hD was timer-controlled for all treatments throughout the experiments. The temperature was maintained at 20–21.5 °C by an 18,000 BTU air condition.

All quantities of 40 ppt water used were first enriched with Walne's medium of nutrients, autoclaved, and according to the desired salinity, were either diluted with

enriched and sterilized distilled water to achieve a salinity of 20 ppt or by the addition of the proper quantity of sterilized artificial salt (Instant Ocean®, Blacksburg, VA, USA) to the desired higher salinities of 50, 60 or 100 ppt. In all the vessels, the suspension of the cells and the supply of $CO_2$ were accomplished using coarse air bubbling through 2-mL glass pipettes (one in every vessel with a supply of half culture volume/min) connected through sterilized plastic hoses to the 0.45 μm filtered central air supply system fed by a blower.

### 2.3. Tested Parameters and Statistical Analysis

The progress of the cultures was monitored by daily measurements of optical density at 750 nm of the medium in each vessel using a visible-UV spectrophotometer (Shimadzu UVmini 1240 UV-visible). This was accomplished by using the proper equation from the calibration curve of the number of cells vs. absorbance using a dense, premeasured (by hematocytometer) culture of each species with serial dilutions and, additionally, more couples of direct cell counts from culture samples taken every 3 days using a Fuchs-Rosenthal hematocytometer.

The maximum specific growth rate (SGR as doublings day$^{-1}$) was estimated during the early exponential (log) phase of the culture's growth curve using the equation [14]:

$$SGR = (\ln C_2 - \ln C_1)/(t_2 - t_1) \tag{1}$$

where $C_1$ and $C_2$ stand for cells/mL at days $t_1$ and $t_2$, respectively ($t_2 > t_1$).

From the above equation, the generation time $t_g$ of the culture was calculated as days required for duplication using the formula [14]:

$$t_g = 0.6931/SGR \tag{2}$$

The calculation of the dry weight was conducted by filtering a known amount of culture through 0.45 μm GF/C filters in a vacuum pump (Heto-SUE-3Q), washing the filter with ammonium formate and drying the filter in an oven to 100 °C for 2 h. Then, the filter was weighed to the fourth decimal and the dry weight was calculated as g/L after subtraction of the pre-weighted filter's tare weight.

The pH was measured daily by a digital pH-meter (HACH-HQ30d-flexi). Statistical treatment of the different variables was performed with ANOVA and the pair-wise Tukey's test for comparison of the means at the 0.05 level of significance using the free PAST3 software.

## 3. Results

### 3.1. Amphidinium carterae

Preliminary experiments have shown that this species (Figure 1) is sensitive to excessive stress caused by intense aeration, so instead of wide Erlenmeyer containers that require intense aeration in stirring the water to achieve uniformity of cell dispersion, which would cause cellular strain, 1-L plastic bottles (Figure 2) were preferred. Thus, with the minimum but sufficient ventilation (flow ~0.3 L/min), the needs of photosynthesis are ensured (continuous exposure of the whole volume of the culture to light) as well as the minimal stress of the cells.

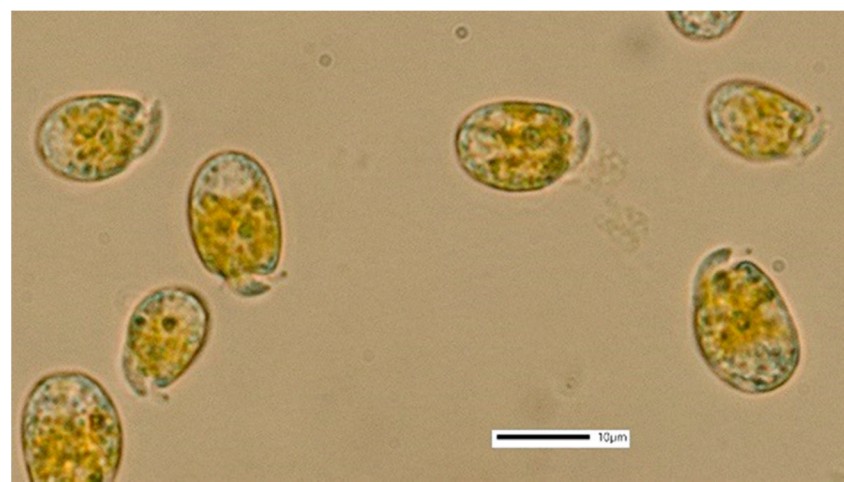

**Figure 1.** The cells of *Amphidinium carterae* at 1000× magnification.

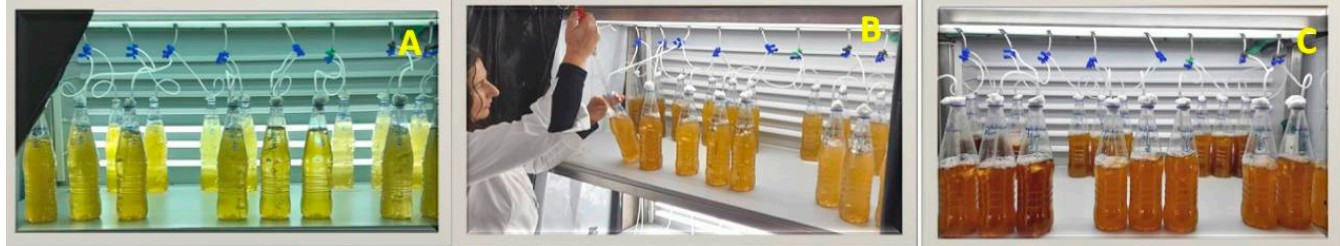

**Figure 2.** The bottles of the culture of *Amphidinium carterae* at differed stages of growth. (**A**): 2nd day, (**B**): 8th day, (**C**): 13th day.

The cultures lasted 15 days, and from the 8th day, the drop of the growth rate and the "entry" in the static (stationary) phase of each culture became visible in all conditions (although in a different way in each). At the salinity of 20 ppt (Figure 3, left), the evolution of the culture showed a much more intense increase in high light (8000 lux) compared to low light (2000 lux). In both lights, there was a very short (2 days) initial phase of adaptation (lag phase) after which, especially in high light, the increase became strongly exponential (log phase). In the high light culture, the final cell density (~6.5 × 10⁶ cells/mL) was almost triple that of the low light (~2.3 × 10⁶ cells/mL). The pH fluctuated in the alkaline region with values 8.3–9.3 and from the beginning it showed higher values in the high light (a sign of more intense photosynthesis) and then (from the 11th day) an abrupt synchronized drop (a sign of aging) in both lights. At salinities of 40 and 50 ppt (Figure 3, middle and right) the increase of the growth curves and the final densities attained after an initial adaptation phase of 2 and 3 days respectively, were much lower compared to 20 ppt, and the stationary phase was reached on the 10th day (on the 14th day at 20 ppt). At the salinity of 40 ppt, the final density in high light was ~3.1 × 10⁶ cells/mL, almost triple that of low light (~1.4 × 10⁶ cells/mL). At the salinity of 50 ppt, even lower values of final densities were recorded in both high (~2.0 × 10⁶ cells/mL) and low light (~0.65 × 10⁶ cells/mL). The pH at these salinities followed the same fluctuation pattern observed in 20 ppt with higher alkaline values (>9.0) in high light as compared to low light (<9.0) and near the end of the culture period, a synchronized drop occurred to almost identical lower values (8.0–8.5).

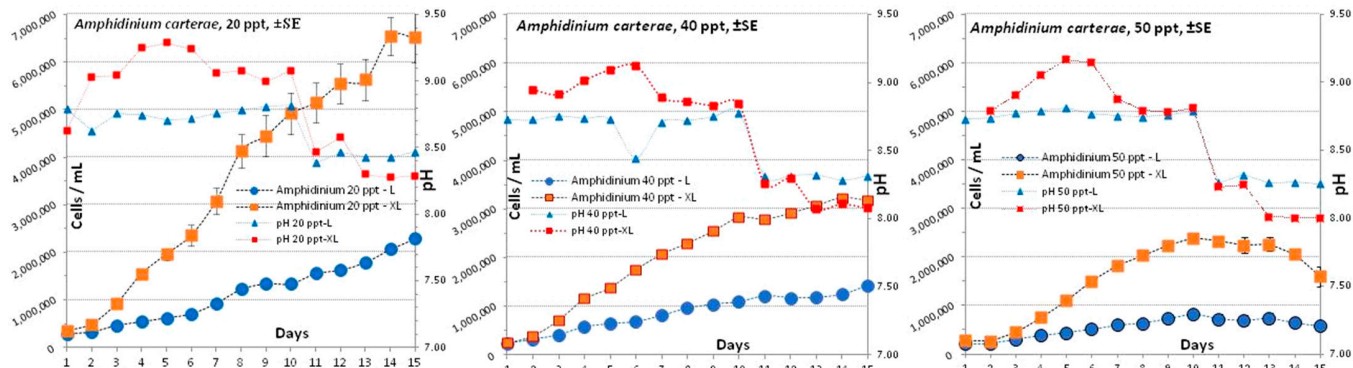

**Figure 3.** The growth curves (in cells/mL) of the culture of *Amphidinium carterae* at salinities of 20 (**left**), 40 (**middle**), and 50 ppt (**right**) and at each light intensity (L: 2000 lux, XL: 8000 lux); also depicted are the pH daily values in each condition.

For the calculation of the specific growth rate (SGR) and the doubling or generation time (tg), the 3nd–8th day interval was chosen as it showed similarity in the upward trend of the growth curve in all salinities. There was a clearly higher growth rate in high light (0.295, 0.232 and 0.302) for salinities 20, 40 and 50 ppt respectively, compared to the values 0.189, 0.167 and 0.143 for low light at the corresponding salinities (Table 1). Statistically, the values differed from each other except those of 20 and 50 ppt, which were statistically equal in high light. As a reflection of the above growth rates, the generation times ($t_g$) were shorter in the high light condition (2.35, 2.98 and 2.29 days for the salinities of 20, 40 and 50 ppt respectively) compared to the values (3.67, 4.16 and 4.85) for low light at the corresponding salinities.

**Table 1.** Data on specific growth rate (SGR) and generation or doubling time ($t_g$) of *Amphidinium carterae* cultures at salinities of 20, 40 and 50 ppt and at each light intensity (L: 2000 lux, XL: 8000 lux).

| Conditions | 20 ppt-L | 20 ppt-XL | 40 ppt-L | 40 ppt-XL | 50 ppt-L | 50 ppt-XL |
|---|---|---|---|---|---|---|
| SGR ± SE | 0.189 [a] ± 0.0077 | 0.295 [b] ± 0.0064 | 0.167 [c] ± 0.0025 | 0.232 [d] ± 0.0022 | 0.143 [e] ± 0.0074 | 0.302 [f,b] ± 0.0023 |
| $t_g$ (days) ± SE | 3.67 ± 0.185 | 2.35 ± 0.049 | 4.16 ± 0.062 | 2.98 ± 0.028 | 4.85 ± 0.323 | 2.29 ± 0.018 |

The period for the calculation of SGR and $t_g$ was from the third to the eighth day. The number of measurements performed totaled 27. The different superscripts (a, b, c, d, e, f) indicate a statistically significant difference at the 0.05 level of confidence (statistical processing with ANOVA and then pair-wise comparison with Tukey's test). Statistically equal values are indicated by repeating the corresponding superscripts.

For the yield of the cultures (Figure 4) in biomass as dry weight per liter of culture (g/L), the values were calculated on the 13th day. It is clearly shown, (in agreement with the growth curves of Figure 2 that concerns the salinity of 20 ppt compared to the curves for the salinities of 40 and 50 ppt respectively), that the salinity of 20 ppt and the high light (8000 lux) causes the highest biomass production (1.19 g/L) with all other conditions giving values below 0.8 g/L. Further noteworthy in all salinities is the much higher production in high light conditions (8000 lux) compared to their counterparts in low light (2000 lux).

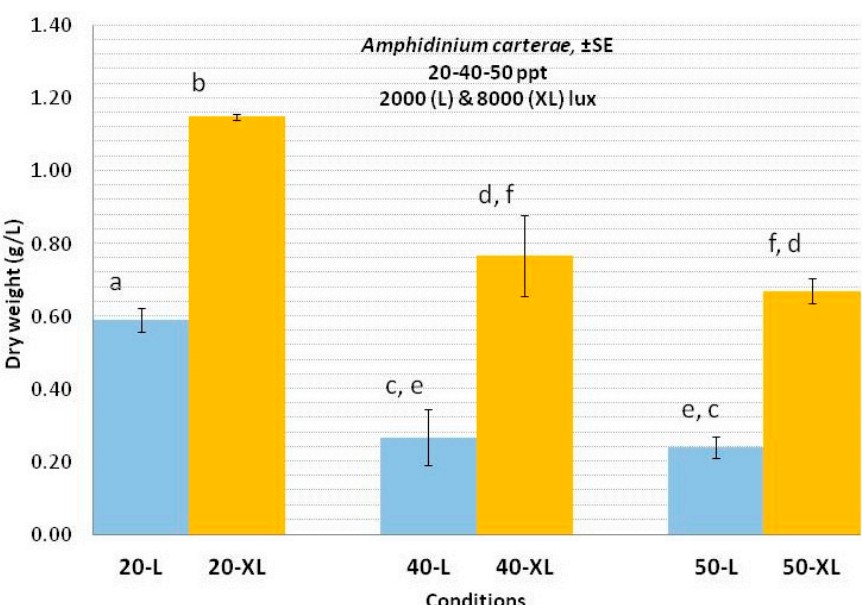

**Figure 4.** Dry weight yield (g/L) ± SE of *Amphidinium carterae* at salinities of 20, 40 and 50 ppt and at each light intensity (L: 2000 lux, XL: 8000 lux). The existence of a statistically significant difference at the 0.05 level is indicated by a different letter. Statistically equal values are indicated by repeating the corresponding indicators on the bars (statistical processing with ANOVA and then pair-wise comparison with Tukey's test).

*3.2. Nephroselmis sp.*

For *Nephroselmis* sp. (Figure 5) cultures, 2-L glass Erlenmeyer conical flasks with two duplicates for each combination of salinity and light intensity were used (Figure 6) and a continuous air supply of half container volume/min (~1 L/min) was provided. The cultures lasted 22 days (Figure 7) and after an initial adaptation-delay period of 3–4 days, all showed an intense and prolonged exponential phase, which was more intense in high light (8000 lux). The "entry" to the static phase of each culture varied in each condition after the 17th day for cultures in high light, while in the low light (2000 lux) at salinities of 20 and 40 ppt, practically for the entire period there was no decrease observed in the exponential growth curve, but it was observed at the salinity of 60 ppt from day 17 (Figure 7, right).

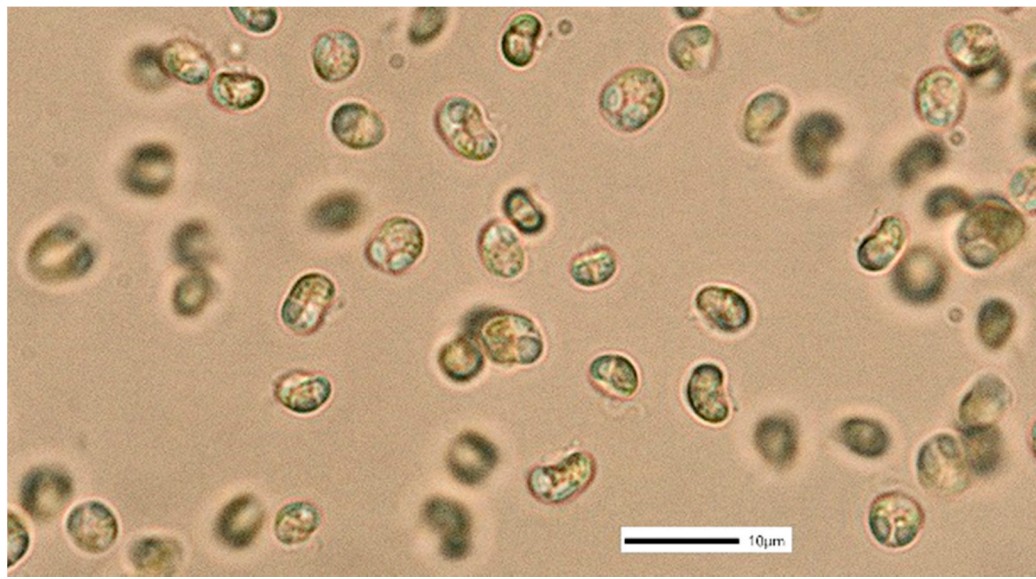

**Figure 5.** The cells of *Nephroselmis* sp. at 1000× magnification.

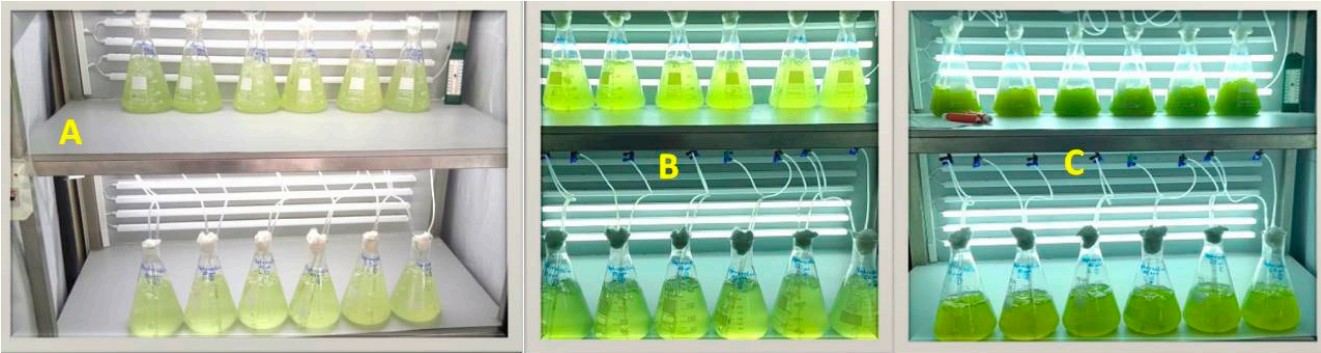

**Figure 6.** The vessels of the culture of *Nephroselmis* sp. at differed stages of growth. (**A**): 1st day, (**B**): 4th day, (**C**): 15th day.

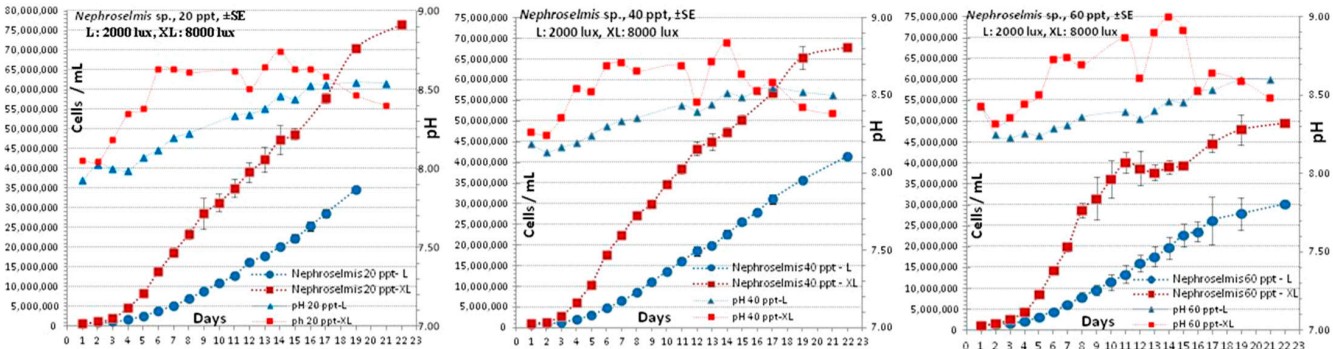

**Figure 7.** The growth curves (in cells/mL) of the culture of *Nephroselmis* sp. at salinities of 20 (**left**), 40 (**middle**) and 60 ppt (**right**) and at each light intensity (L: 2000 lux, XL: 8000 lux); also depicted are the pH daily values in each condition.

At 20 ppt salinity (Figure 7, left) the evolution of the culture was evident, which showed a much more intense increase in high light (8000 lux) reaching densities of about $75 \times 10^6$ cells/mL on the 22nd day compared to the low light (2000 lux) where on the 20th day it reached $35 \times 10^6$ cells/mL ($70 \times 10^6$ cells/mL that day in high light).

The pH fluctuated at the alkaline level with initial values of ~7.9, which quickly rose to the level of ~8.5, and from the beginning already showed higher values in high light (a sign of more intense photosynthesis) and then (from day 17) a synchronized drop (a sign of aging) in both lights. At 40 ppt salinity (Figure 7, middle), the culture also showed a much more intense increase in high light (8000 lux) reaching a density of about $67 \times 10^6$ cells/mL on the 22nd day compared to the low light (2000 lux), which reached $41 \times 10^6$ cells/mL on the same day. In both light intensities, after a very short initial phase (3–4 days) of delay-adaptation, the increase became strongly exponential, especially in high light. The pH fluctuated in the alkaline level with values of ~8.2–8.8 and from the beginning it already showed higher values in high light and then (from the 17th day) a synchronized drop in both light intensities. At the salinity of 60 ppt (Figure 7, right) the culture also showed a much more intense increase in high light (8000 lux) reaching a density of about $50 \times 10^6$ cells/mL on the 22nd day compared to the low light (2000 lux), which reached $30 \times 10^6$ cells/mL on the same day. In both light intensities, after a very short initial phase (3–4 days) of delay-adaptation, the increase became strongly exponential especially in high light but lasted only until the 11th day while in the low light it lasted up to the 17th day. The cultures then entered the static phase, which was most evident in high light.

For the calculation of the specific growth rate (SGR) and the doubling or generation time ($t_g$), the 4th–8th day interval was chosen as it showed similarity in the upward trend of the growth curve in all salinities. There was a clearly higher growth rate in all salinities with high light (0.401, 0.370 and 0.462) for salinities 20, 40 and 60 ppt respectively, compared to the values (0.338, 0.341 and 0.336) for low light at the corresponding salinities (Table 2). Statistically, the values differed from each other except those of 20, 40 and 60 ppt, which in

low light were statistically equal. As a reflection of the above growth rates, the generation times ($t_g$) were shorter in the high light condition (1.729, 1.875 and 1.499 days for the salinities of 20, 40 and 60 ppt respectively) compared to the values (2.051, 2.032 and 2.063) for low light at the corresponding salinities.

**Table 2.** Data on specific growth rate (SGR) and generation or doubling time ($t_g$) of *Nephroselmis* sp. cultures at salinities of 20, 40 and 60 ppt and at each light intensity (L: 2000 lux, XL: 8000 lux).

| Conditions | 20 ppt-L | 20 ppt-XL | 40 ppt-L | 40 ppt-XL | 60 ppt-L | 60 ppt-XL |
|---|---|---|---|---|---|---|
| SGR ± SE | 0.338 [a] ± 0.001 | 0.401 [b] ± 0.0021 | 0.341 [c,a,e] ± 0.0016 | 0.370 [d] ± 0.0006 | 0.336 [e,a,c] ± 0.0008 | 0.462 [f] ± 0.0023 |
| $t_g$ (days) ± SE | 2.051 ± 0.003 | 1.729 ± 0.009 | 2.032 ± 0.01 | 1.875 ± 0.003 | 2.063 ± 0.005 | 1.499 ± 0.011 |

The period for the calculation of SGR and $t_g$ was from the fourth to the eighth day. The number of measurements totaled 18. The different superscripts (a, b, c, d, e, f) indicate a statistically significant difference at the 0.05 level of confidence (statistical processing with ANOVA and then pair-wise comparison with Tukey's test). Statistically equal values are indicated by repeating the corresponding superscripts.

For the yield of the cultures (Figure 8) in biomass as dry weight per liter of culture (g/L), the values were calculated on the 17th day. It is clearly shown, in agreement with the increase of the curves in Figure 7, that the salinities of 20 and 40 ppt resulted in a much higher yield (2.88 and 2.99 g/L respectively) compared to the salinity of 60 ppt (1.61 g/L) in high light (8000 lux). In all salinities in low light, the yields were much lower, around 1 g/L, and statistically equal.

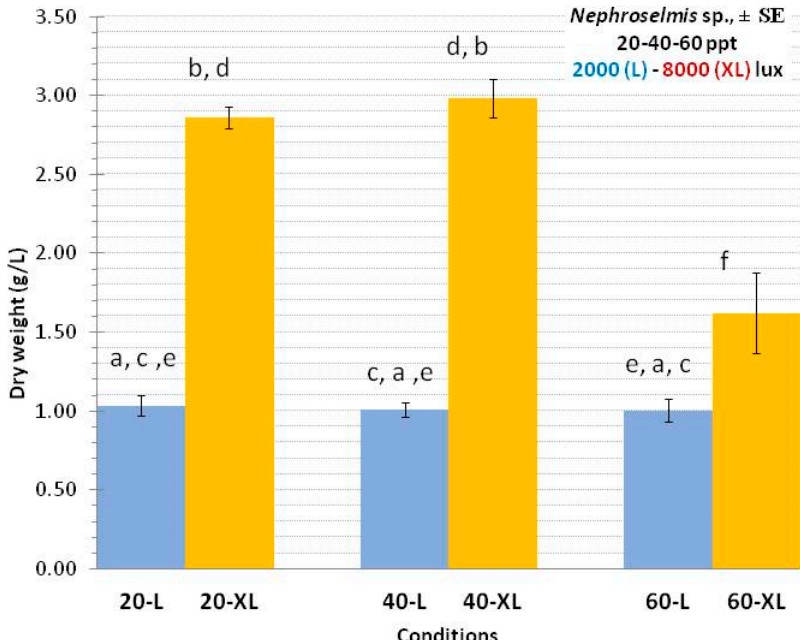

**Figure 8.** Dry weight yield (g/L) ± SE of *Nephroselmis* sp. at salinities of 20, 40 and 60 ppt and at each light intensity (L: 2000 lux, XL: 8000 lux). The existence of a statistically significant difference at the 0.05 level is indicated by a different letter. Statistically equal values are indicated by repeating the corresponding indicators on the bars (statistical processing with ANOVA and then pair-wise comparison with Tukey's test).

### 3.3. Tetraselmis sp. (var. red pappas)

The cultures of *Tetraselmis* sp. (var. red pappas) (Figure 9) in all salinities (20, 40 and 60 ppt) were conducted in 2-L glass conical Erlenmeyer flasks (Figure 10) receiving 1L of air/min with three replicates for each combination of salinity and light intensity, low and high (2000 and 8000 lux respectively). The culture period lasted 17 days (Figure 11), and in all treatments, after an initial delay-adaptation period of 2–4 days, an intense and continuous exponential growth, much more pronounced in the high light, was observed. The only culture that entered the stationary phase was that of 60 ppt-high light, which



resulted in substantially lower final densities in both high and low light ($5.8 \times 10^6$ and $2.2 \times 10^6$ cells/mL respectively) compared to 20 ppt ($10 \times 10^6$ and $4.3 \times 10^6$ cells/mL respectively) and 40 ppt ($9.6 \times 10^6$ and $3.3 \times 10^6$ cells/mL respectively).

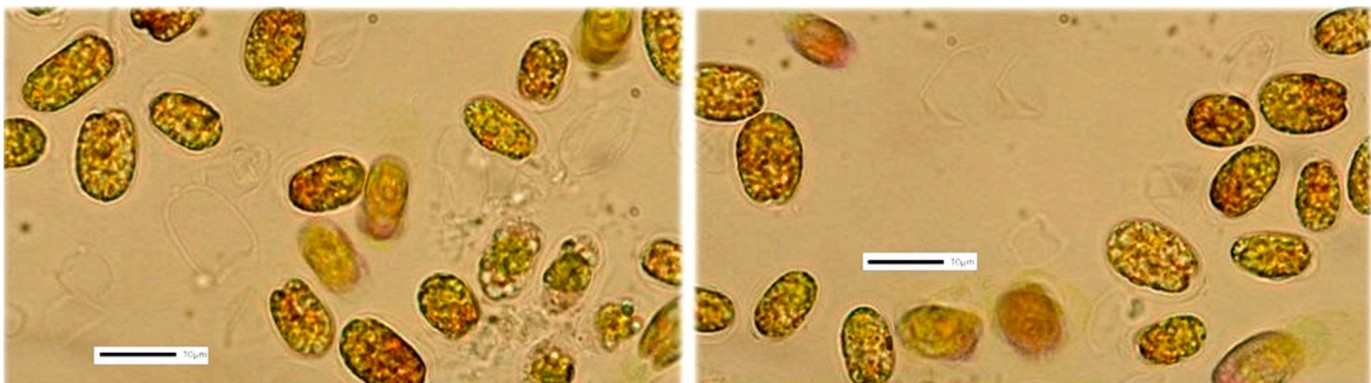

**Figure 9.** The cells of *Tetraslmis* sp. (var. red pappas) at 1000× magnification. Many cells became reddish.

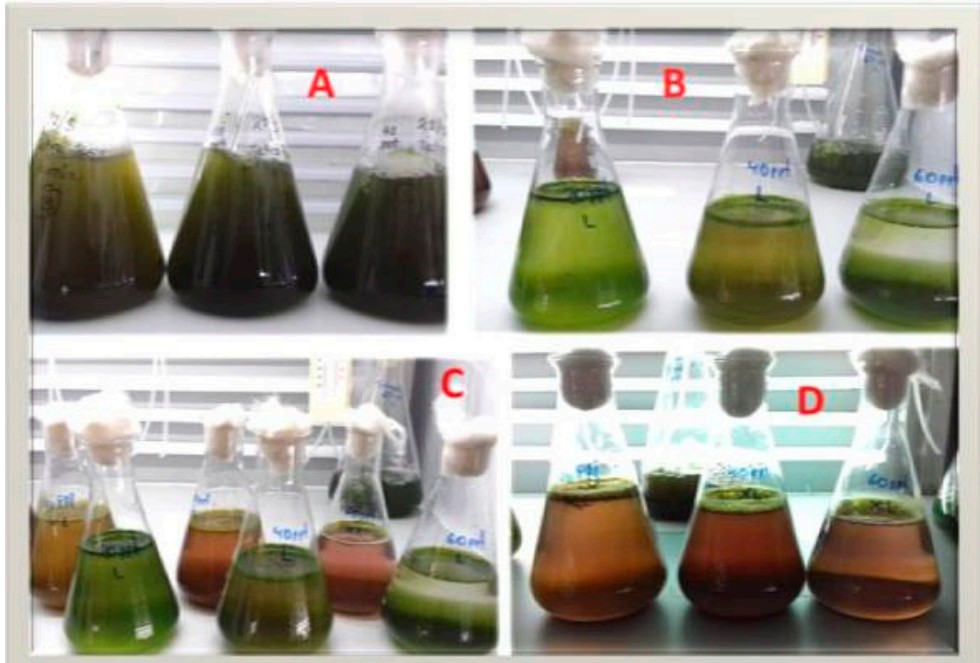

**Figure 10.** The culture vessels of *Tetraselmis* sp. (var. red pappas) in various phases of color change. (**A**): 15th day, cultures with aeration, the color has already become dull dark green. (**B**): 16th day, cultures at rest, many cells have precipitated, the color begins to take on a reddish tinge. (**C**): 17th day, cultures without aeration, the front row comes from light of 2000 lux, the back row from 8000 lux and is clearly reddened. (**D**): 17th day, cultures of 20, 40 and 60 ppt (from left to right) and from bright light (8000 lux) without aeration with their reddish color clearly visible.

Nevertheless, the specific growth rate (SGR) calculated for the period from the third to the ninth day for all treatments did not exhibit a huge difference between all salinities at high light (Table 3) with values higher than 0.3, while in low light, it was much lower at all salinities (0.18–0.203). It seems that high-intensity light has a profound positive affect on the speed of multiplication, enhancing the overall metabolism of the cells.

The fluctuation of pH over the culture period is indicative of more intense photosynthetic activity in the high light cultures in all salinities as it remained substantially over 8.5 while the low light cultures remained at 8.0 or lower.

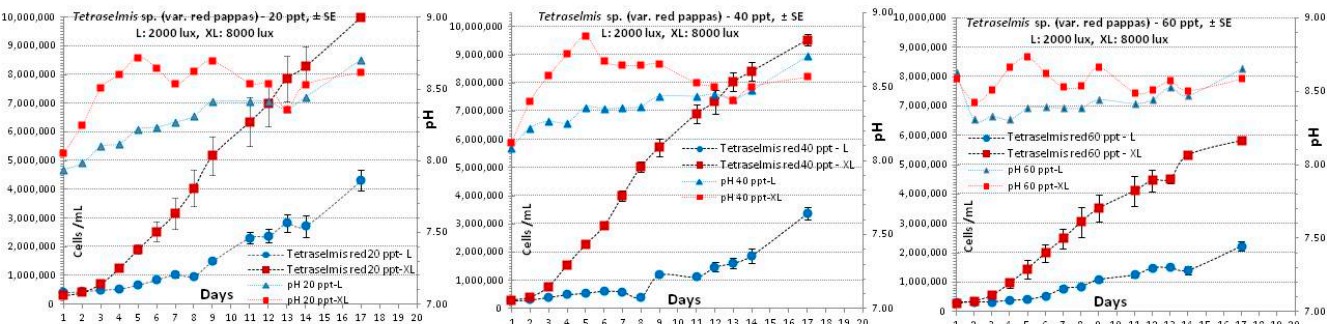

**Figure 11.** The growth curves (in cells/mL) of the culture of *Tetraselmis* sp. (var. red pappas) at salinities of 20 (**left**), 40 (**middle**) and 60 ppt (**right**) and at each light intensity (L: 2000 lux, XL: 8000 lux); also depicted are the pH daily values in each condition.

**Table 3.** Data on specific growth rate (SGR) and generation or doubling time ($t_g$) of *Tetraselmis* sp. (var. red pappas) cultures at salinities of 20, 40 and 60 ppt and at each light intensity (L: 2000 lux, XL: 8000 lux).

| Conditions | 20 ppt-L | 20 ppt-XL | 40 ppt-L | 40 ppt-XL | 60 ppt-L | 60 ppt-XL |
|---|---|---|---|---|---|---|
| **SGR** $\pm$ SE | **0.181** [a] $\pm$ 0.0035 | **0.323** [b] $\pm$ 0.0069 | **0.180** [c,a] $\pm$ 0.0023 | **0.331** [d,b] $\pm$ 0.0037 | **0.203** [e] $\pm$ 0.0031 | **0.310** [f] $\pm$ 0.0019 |
| **$t_g$ (days)** $\pm$ SE | **3.860** $\pm$ 0.074 | **2.163** $\pm$ 0.046 | **3.870** $\pm$ 0.052 | **2.097** $\pm$ 0.023 | **3.429** $\pm$ 0.052 | **2.236** $\pm$ 0.013 |

The period for the calculation of SGR and $t_g$ was from the third to the ninth day. The number of measurements totaled 18. The different superscripts (a, b, c, d, e, f) indicate a statistically significant difference at the 0.05 level of confidence (statistical processing with ANOVA and then pair-wise comparison with Tukey's test). Statistically equal values are indicated by repeating the corresponding superscripts.

For the yield of cultures (Figure 12) in biomass as dry weight per liter of culture (g/L), the values were calculated on the 17th day. It is clearly shown, in agreement with the increase of the curves in Figure 11, that in high light (8000 lux) the salinities of 20 and 40 ppt resulted in much higher yield (2.0 and 1.8 g/L respectively and statistically equal) compared to the salinity of 60 ppt (0.92 g/L). In all salinities at low light, the yields were much lower, around 0.9 g/L, and statistically equal between 20 and 40 ppt and also equal to the relevant one of 60 ppt-XL. The lowest yield (0.78 g/L) was recorded in 60 ppt-L.

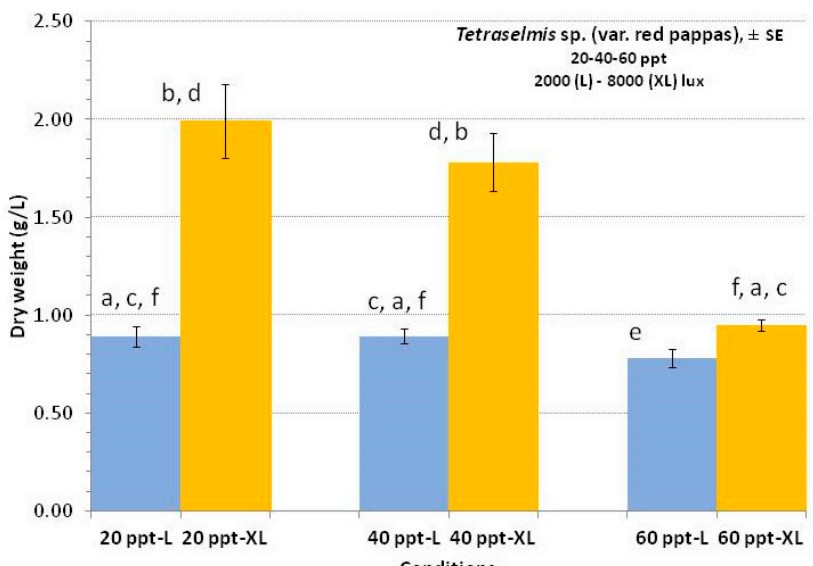

**Figure 12.** Dry weight yield (g/L) $\pm$ SE of *Tetraselmis* sp. (var. red pappas) at salinities of 20, 40 and 60 ppt and at each light intensity (L: 2000 lux, XL: 8000 lux). The existence of a statistically significant difference at the 0.05 level is indicated by a different letter. Statistically equal values are indicated by repeating the corresponding indicators on the bars (statistical processing with ANOVA and then pair-wise comparison with Tukey's test).

### 3.4. Asteromonas gracilis

The case of the chlorophyte *Asteromonas gracilis* (Figure 13) is very special as its natural habitat is exclusively ultra-salty areas in many parts of the Earth. It is a species that can, and prefers to, grow in salinities much higher than seawater. In our study, we isolated it from the saltern basins of the Messolonghi lagoon and it was easily maintained at a salinity of about 100 ppt [15]. From the literature [16] and from preliminary tests it was found that this species does not grow at all in salinities around 20 ppt, so our cultures were formed at 40, 60 and 100 ppt.

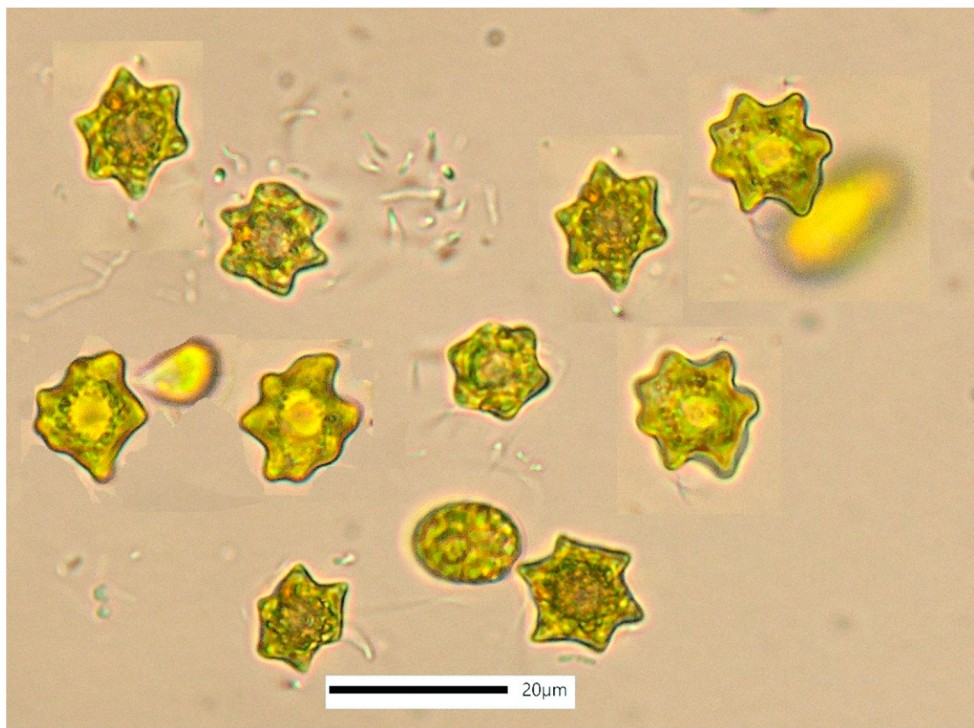

**Figure 13.** The cells of *Asteromonas gracilis* with the majority of them in a star-like appearance.

The cultures of *A. gracilis* (Figure 14) in all salinities (40, 60 and 100 ppt) were conducted in triplicate with two replicates of 2-L glass conical Erlenmeyer flasks receiving ~1 L of air/min and one of a 1-L plastic cylindrical vessel receiving ~0.5 L of air/min for each combination of salinity and light intensity (low-L and high-XL, 2000 and 8000 lux respectively). The culture period lasted 17 days until the cultures had just entered the stationary phase.

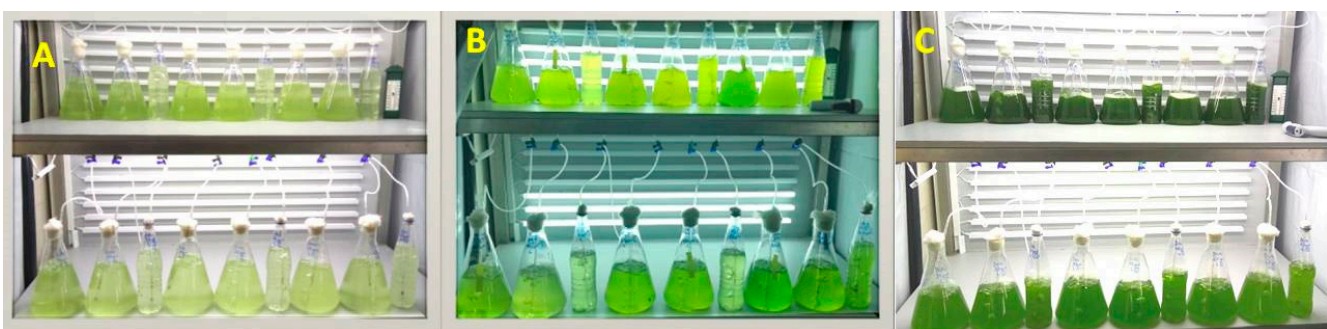

**Figure 14.** The bottles of the culture of *Asteromonas gracilis* at different stages of growth. (**A**): 1st day, (**B**): 4th day, (**C**): 11th day.

The most prominent characteristic of the cultures of *A. gracilis* in all salinities was the rather long initial lag phase that lasted 5–7 days in the lower salinities (40 and 60 ppt), absolutely in a similar manner for both high and low light, compared to 100 ppt that lasted only 2 days for high light and 4 days for low light (Figure 15).

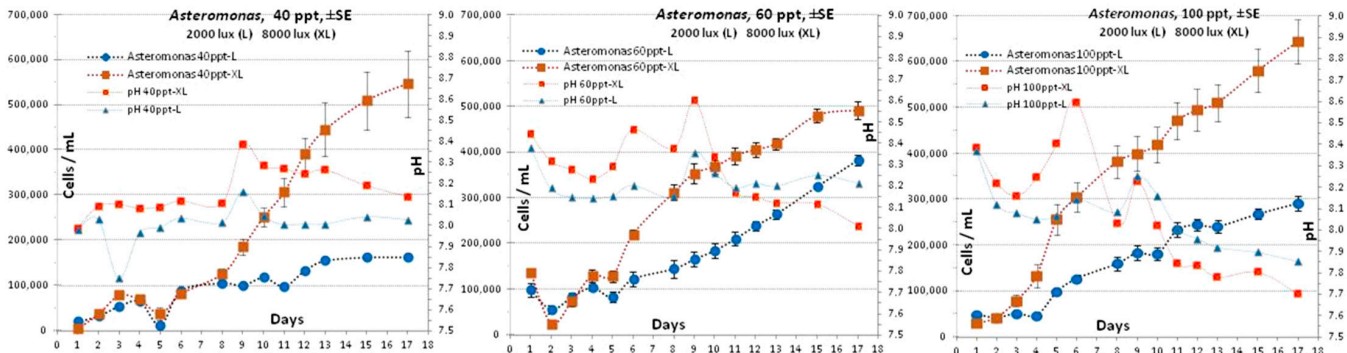

**Figure 15.** The growth curves (in cells/mL) of the culture of *Asteromonas gracilis* at salinities of 40 (**left**), 60 (**middle**), and 100 ppt (**right**) and at each light intensity (L: 2000 lux, XL: 8000 lux); also depicted are the pH daily values in each condition.

The highest value of cell density was recorded in the salinity of 100 ppt at high light on the 17th day ($6.4 \times 10^5$ cells/mL) and was clearly higher than the relevant values of the 60 and 40 ppt at high light ($4.9 \times 10^5$ and $5.5 \times 10^5$ cells/mL respectively). It is worth noting, however, that these maximum values in the salinities of 60 and 40 ppt can be considered as equivalent due to the high amount of variation in the values of 40 ppt as evidenced by the width of the error bars in Figure 15 (left). This was further certified as the pace of growth in high light shown in all three diagrams of Figure 15 presented uniformity until at least the 12th day when cell densities were identical at around $4 \times 10^5$ cells/mL for both 40 and 60 ppt cultures.

The maximum densities in the conditions of low light were much lower compared to high light in all salinities with the maximum value recorded at 60 ppt ($3.9 \times 10^5$ cells/mL) and the lowest at 40 ppt ($1.8 \times 10^5$ cells/mL) and $2.9 \times 10^5$ cells/mL at 100 ppt.

The values of pH fluctuated greatly in all treatments in the region of 8.0–8.5 during the whole culture period for the cultures of 40 and 60 ppt, with higher values steadily recorded at high light over those of low light but after the 11th day, the values of high light decreased substantially and in the salinity of 60 ppt dropped below those of low light. This phenomenon was much exaggerated in the salinity of 100 ppt where the drop started from the ninth day and was additionally manifested for both light intensities, with values plummeted to below 8.0 (Figure 15, right).

The specific growth rate (SGR) was calculated for the period from the third to the ninth day based on the findings of the growth curves of Figure 15 in order to obtain values containing on the one hand, a part of the initial lag phase (third day) and on the other, a representative day of the exponential (log) phase around its middle period (ninth day). The highest growth rate of 0.280 doublings/day was recorded in 100 ppt-XL and in 60 ppt-XL (0.273), which were statistically equal (Table 4), followed by 100 ppt-L (0.205) and 60 ppt-L (0.190), while the lowest salinity of 40 ppt resulted in the lowest values (0.143 and 0.106) for both high (XL) and low light (L) respectively. As of these values, the shortest generation time ($t_g$) of 2.551 days was recorded in 100 ppt-XL and the longest (8.034 days) in 40 ppt-L.

The yield in g dry weight/L was measured at the final day of the culture period (17th day) and from the very first glance it becomes evident that the high light condition resulted in much higher production of biomass compared to its low light counterparts in each salinity. The maximum value (0.70 g/L) was recorded in the treatment of 100 ppt-XL and the lowest (0.25 g/L) in 40 ppt-L (Figure 16). Worth noting are the statistically equal values of both 40 ppt-XL and 60 ppt-XL (0.52 and 0.51 g/L respectively) and also those of 40 ppt-L and 60 ppt-L (0.38 and 0.36 g/L respectively).

**Table 4.** Data on specific growth rate (SGR) and generation or doubling time ($t_g$) of *Asteromonas gracilis* cultures at salinities of 40, 60 and 100 ppt and at each light intensity (L: 2000 lux, XL: 8000 lux).

| Conditions | 40 ppt-L | 40 ppt-XL | 60 ppt-L | 60 ppt-XL | 100 ppt-L | 100 ppt-XL |
|---|---|---|---|---|---|---|
| SGR $\pm$ SE | 0.106 [a] $\pm$ 0.007 | 0.143 [b] $\pm$ 0.01 | 0.190 [c,e] $\pm$ 0.005 | 0.273 [d,f] $\pm$ 0.014 | 0.205 [e] $\pm$ 0.009 | 0.280 [f] $\pm$ 0.01 |
| $t_g$ (days) $\pm$ SE | 8.034 $\pm$ 0.773 | 5.674 $\pm$ 0.426 | 3.711 $\pm$ 0.101 | 2.768 $\pm$ 0.154 | 3.607 $\pm$ 0.175 | 2.551 $\pm$ 0.07 |

The period for the calculation of SGR and $t_g$ was from the third to the ninth day. The number of measurements totaled 30. The different superscripts (a, b, c, d, e, f) indicate a statistically significant difference at the 0.05 level of confidence (statistical processing with ANOVA and then pair-wise comparison with Tukey's test). Statistically equal values are indicated by repeating the corresponding superscripts.

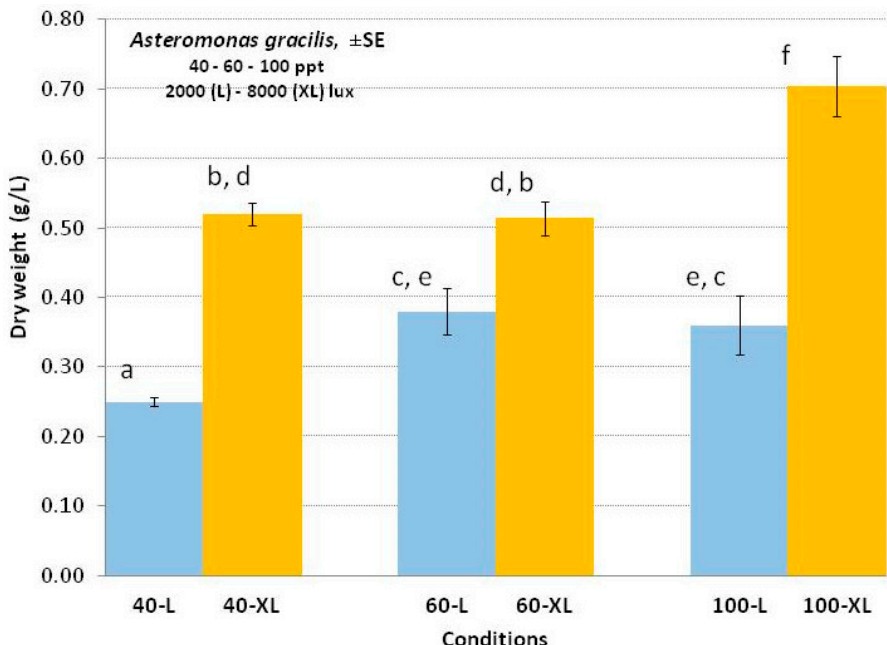

**Figure 16.** Dry weight yield (g/L) $\pm$ SE of *Asteromonas gracilis* at salinities of 40, 60 and 100 ppt and at each light intensity (L: 2000 lux, XL: 8000 lux). The existence of a statistically significant difference at the 0.05 level is indicated by a different letter. Statistically equal values are indicated by repeating the corresponding indicators on the bars (statistical processing with ANOVA and then pair-wise comparison with Tukey's test).

*3.5. Dunaliella* sp.

The cultures of *Dunaliella* sp. (Figure 17) in all salinities (20, 40 and 60 ppt) were conducted in triplicate (Figure 18) with two replicates of 2-L glass conical Erlenmeyer flasks receiving ~1 L of air/min and one of a 1-L plastic cylindrical vessel receiving ~0.5 L of air/min for each combination of salinity and light intensity (low-L and high-XL, 2000 and 8000 lux, respectively). The culture period lasted 22 days as all cultures had already entered the stationary phase around the 14th day.

In all treatments (Figure 19), the lag phase was very short (2–3 days) after which a sharp elevation of the growth curve characterized the log phase, especially in high light of the 40 and 60 ppt salinities in contrast to the low light curves where the transition to the log phase was long and smooth.

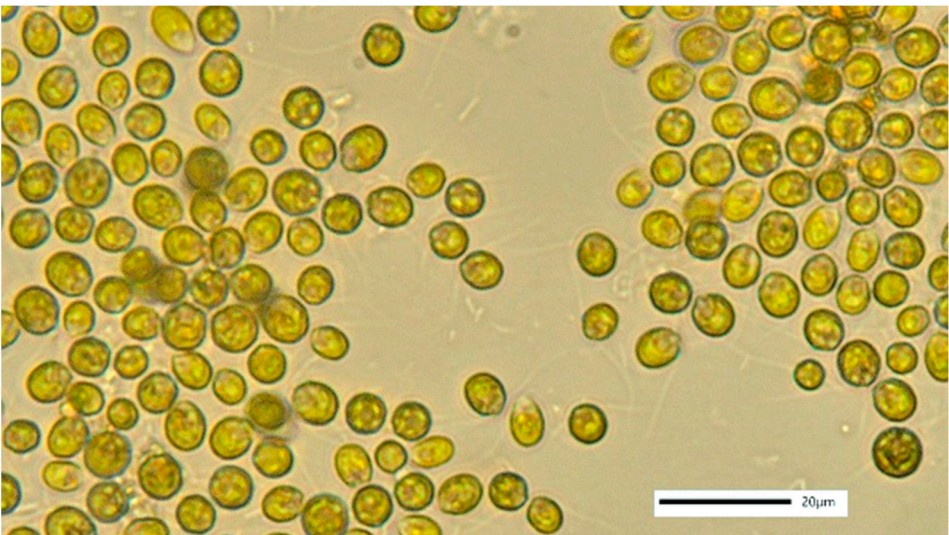

**Figure 17.** The cells of *Dunaliella* sp. at 630× magnification.

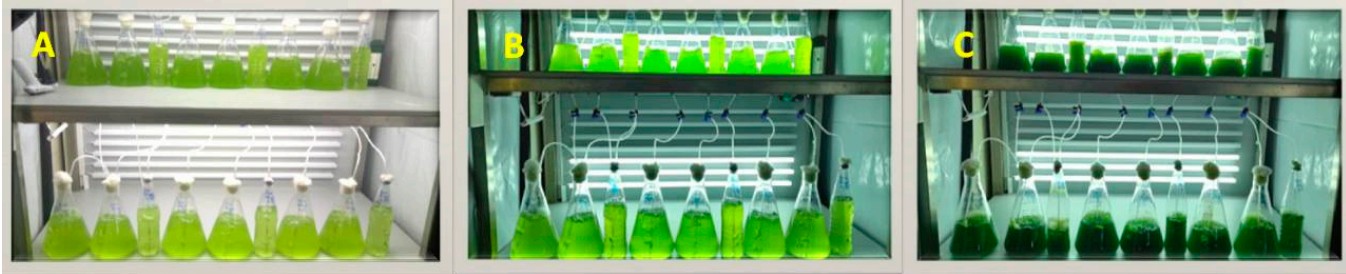

**Figure 18.** The bottles of the culture of *Dunaliella* sp. at different stages of growth. (**A**): 1st day, (**B**): 4th day, (**C**): 11th day.

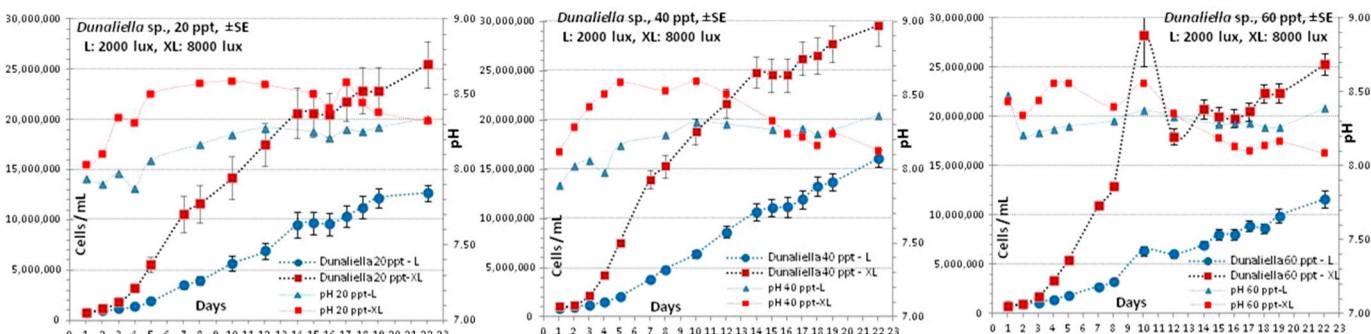

**Figure 19.** The growth curves (in cells/mL) of the culture of *Dunaliella* sp. at salinities of 20 (**left**), 40 (**middle**) and 60 ppt (**right**) and at each light intensity (L: 2000 lux, XL: 8000 lux); also depicted are the pH daily values in each condition.

In all salinities at high light, cell densities measured on the 22nd day, recorded values over $25 \times 10^6$ cells/mL while their counterparts at low light fluctuated around half that value. The maximum densities were recorded in the salinity of 40 ppt with $29 \times 10^6$ cells/mL and $16 \times 10^6$ cells/mL for high and low light, respectively.

pH fluctuated intensively in the salinities of 20 and 40 ppt with values around 7.9 at the start and rising to 8.6 in high light where it stabilized for most of the culture period, while the values for low light remained substantially lower in the salinity of 20 ppt. Only after the 13th day were the pH values of both light intensities equated until the termination of the experiment. In the salinity of 60 ppt, the pattern of pH fluctuation was similar to those of 20 and 40 ppt with the only difference being the drop in values for high light below those for low light after the middle of the culture period.

The specific growth rate (SGR) calculated for the period from the third to the eighth day was based on the uniformity of the shape of the growth curves of all treatments in Figure 19. The highest growth rate of 0.405 doublings/day was recorded in 60 ppt-XL and in 40 ppt-XL (0.387), which were statistically equal (Table 5) followed by 20 ppt-XL (0.350). The values at low light in all salinities were substantially lower (0.215–0.279). As of these values, the shortest generation time ($t_g$) of 1.77 days was recorded in 60 ppt-XL and the longest (3.24 days) in 60 ppt-L.

**Table 5.** Data on specific growth rate (SGR) and generation or doubling time ($t_g$) of *Dunaliella* sp. cultures at salinities of 20, 40 and 60 ppt and at each light intensity (L: 2000 lux, XL: 8000 lux).

| Conditions | 20 ppt-L | 20 ppt-XL | 40 ppt-L | 40 ppt-XL | 60 ppt-L | 60 ppt-XL |
|---|---|---|---|---|---|---|
| SGR | **0.239** [a] ± 0.01 | **0.350** [b] ± 0.015 | **0.279** [c,a] ± 0.004 | **0.387** [d,b] ± 0.0037 | **0.215** [e] ± 0.003 | **0.405** [f,d] ± 0.007 |
| $t_g$ (days) | **3.020** ± 0.111 | **2.09** ± 0.095 | **2.495** ± 0.036 | **1.815** ± 0.042 | **3.246** ± 0.052 | **1.774** ± 0.027 |

The period for the calculation of SGR and $t_g$ was from the third to the eighth day. The number of measurements totaled 27. The different superscripts (a, b, c, d, e, f) indicate a statistically significant difference at the 0.05 level of confidence (statistical processing with ANOVA and then pair-wise comparison with Tukey's test). Statistically equal values are indicated by repeating the corresponding superscripts.

The yield in g dry weight/L was measured on the final day of the culture period (22nd day) and from the very first glance it is evident that the high light condition resulted in much higher production of biomass compared to the low light counterparts in each salinity. The maximum values (1.58 and 1.56 g/L) were recorded in the treatments of 40 and 60 ppt-XL respectively, and the lowest (0.59 and 0.45 g/L) in 20 and 60 ppt-L respectively (Figure 20).

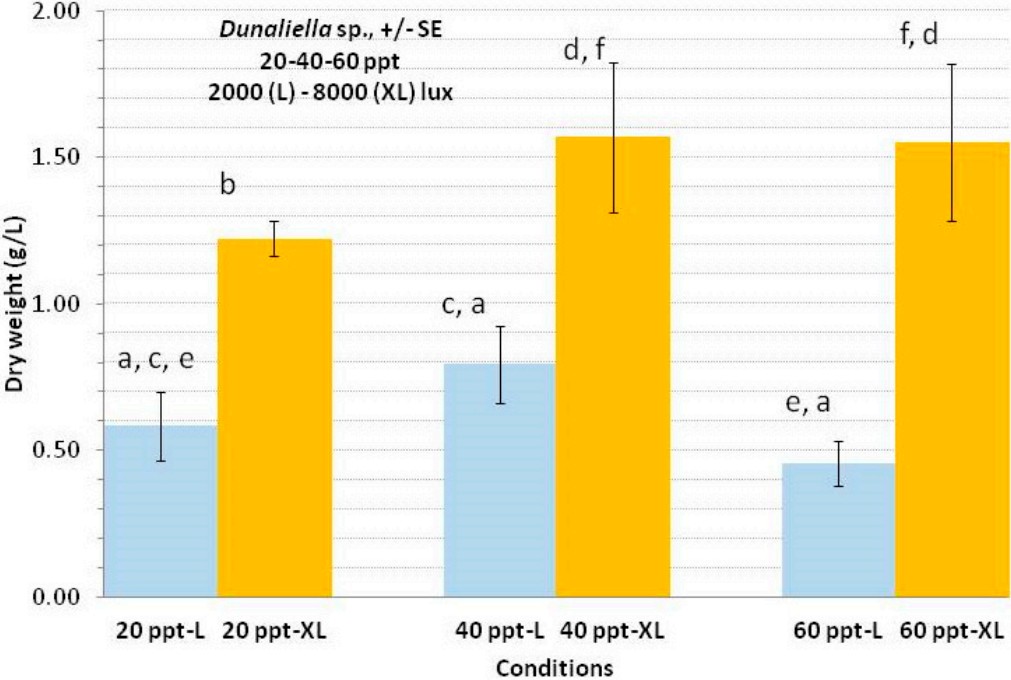

**Figure 20.** Dry weight yield (g/L) ± SE of *Dunaliella* sp. at salinities of 20, 40 and 60 ppt and at each light intensity (L: 2000 lux, XL: 8000 lux). The existence of a statistically significant difference at the 0.05 level is indicated by different letters. Statistically equal values are indicated by repeating the corresponding indicators on the bars (statistical processing with ANOVA and then pair-wise comparison with Tukey's test).

## 4. Discussion

A considerable bulk of data exists in the literature that concerns the effects of various manipulations of physicochemical parameters on the growth and biochemical output of

microalgae cultures. Two major drawbacks are recognizable on reviewing this issue in the literature. First, the immense variation in the tools (e.g., vessels) and techniques (e.g., batch or continuous culture, nutrients, light, temperature, etc.) used. The second drawback is the fragmented information that the results of each work transfer to the reader. An example of this is the calculation of the specific growth rate (SGR), which in the majority of papers is presented with no reference of the exact culture time on which it was calculated upon. This can cause great discrepancies of the calculated values and comparisons are greatly hindered. In our study, we calculated SGR from around the end of the lag (stationary) phase until the middle of the log (exponential) phase. Had it been otherwise (i.e., from the well-advanced log phase), our values would have been much higher. This resulted in a more realistic approach to how each particular tested species is presented in reality. It makes little sense to record SGR only during the short log phase whilst a long lag phase has preceded it, e.g., [17].

It is logical and essential to rank the priorities on the path of investigating and understanding every species needs and potentials in culture terms. With no doubt, the first priority is to investigate the maximum possible biomass that can be produced under economically feasible conditions. For this purpose, a kind of default set of parameters should be implemented in order to be considered as a starting point. In the present paper, an ordinary indoor temperature of 20–21 °C, a classic nutrient medium (Walne), a sufficient range of illumination (2000–8000 lux), bubble aeration with no addition of $CO_2$, vessels of 1–2 L that are not too small nor too big, and non-manipulated pH were chosen for our purpose, which was to investigate the evolution of the growth curve, the growth rate, and the dry weight yield of each species in salinities of brackish, sea, and hypersaline waters. Only after gathering results from experimentation similar to the abovementioned conditions can the procedures for further manipulations in order to achieve maximum biochemical products (e.g., pigments, lipids, etc.) have meaning. It is of little use to find that a certain species produces a great deal of valuable substances in conditions that otherwise result in minimum biomass from a very small growth rate and a very long culture period.

Of the five species of microalgae cultured in the present paper, only one is represented quite well in terms of Genus (*Dunaliella* sp.) in the relevant literature, followed by *Tetraselmis*. *Amphidinium* has only limitedly been studied, and even less so *Nephroselmis*. *Asteromonas gracilis* is almost neglected completely, although its importance as feed for the rotifer *Brachionus plicatilis* is well documented [18,19] and also presents other advantageous features [15].

*4.1. Amphidinium carterae*

The interest in the culture of this species and of the dinoflagellates in general is based on their capacity to produce bioactive metabolites (amphidinolides and amphidinols) and possibly biofuels, although there are concerns about their ability to withstand shear stress caused by turbulence in photobioreactors [20–23]. From the existing limited data concerning the effects of light and salinity on its culture growth, no solid conclusions can be drawn, as either the vessels used were extremely different, ranging from minute 25 mL [24] to small (200 mL, [25]), to medium (1 L, [26]), to Photobioreactors [17]), or the methods applied in the calculation of specific growth rate were not fully clarified. Only the optimum salinity for the best growth can be indirectly set with certainty at a value of less than 35 ppt, which was the case in our study and was documented [24] for another relative species (*Amphidinium klebsii*). The very low yield in our work both in density of cells and dry biomass (g/L) in our 40 and 60 ppt cultures as compared to the respective ones in 20 ppt, strongly indicates that *A. carterae* responds much better in lower salinities. Concerning the effect of light, the situation is even more complicated in the literature as it is not only the various light intensities used and their units (e.g., what is 4000 µW cm$^{-2}$ [20] and how can this can be transformed to lux based on the well-known formula: 5 µmol photons m$^{-2}$ s$^{-1}$ = 250 lux = 1 watt m$^{-2}$?), but also the various temperatures used, either much higher than ours (e.g., 25–30 °C, [26]) or similar (~20 °C) to ours [17,24], the addition

of $CO_2$ [25], and a puzzling combination of them and others, when looked upon trying to decipher the existing topic for the optimum conditions. We observed far better growth in the cultures of *A. carterae* in high light intensity (8000 lux) compared to low light (2000 lux) and this occurred in all salinities tested. The same was documented in all previous studies, but as yields are highly variable in them, we consider our results of the growth rate and cell density (SGR = ~0.3 at high light, ~0.18 at low light and $6.5 \times 10^6$ cells/mL and $2.3 \times 10^6$ cells/mL respectively) and yield (1.19 g/L at 20 ppt-XL vs. 0.59 g/L at 20 ppt-L) far more realistic than those of [17] (0.473, 0.226 g/L, and $69 \times 10^3$ cells/mL with more than 10 days lag phase at 2000 lux) or [26] (0.41 and $100 \times 10^3$ cells/mL at ~3000 lux).

*4.2. Nephroselmis sp.*

The chlorophyte *Nephroselmis* sp. has drawn interest due to its potential for the production of lipids, carotenoids, and various antioxidants [27–29] and as an excellent feed for *Artemia* [30]. Its documented mixotrophic ability [29] is an advantage for its mass culture as it can occasionally overcome the shortage of nutrients and even benefit from the bacterial load usually present in old cultures. Because of its special ability for mixotrophy, the cultures of *Nephroselmis* almost never collapse (personal observation) and remain alive for many weeks in the stationary phase, subjected only to slight discoloration (fade green) in comparison to its exponential (log) phase color (green). After the renewal of the medium with enriched water, the growth and the color are revived. From the outcome of our cultures, it is shown that it is highly productive both in terms of final cell density and yield in dry weight. It grows well in a wide range of salinities from half- to full-strength seawater and reaches densities in the order of ~70 x $10^6$ cells/mL and yields close to 3.0 g d.w./L in about 20 days provided that light intensity is at least 8000 lux. Both were attained in the batch culture at a temperature of ~20 °C, with no addition of extra $CO_2$, and compared to the relevant values of [28] (0.5 g/L at 25 °C, 6000 lux, and addition of $CO_2$) or [27] (17–30 $\times 10^6$ cells/mL, 0.35–0.56 g/L at 26.5 °C, and continuous illumination of 30,000 lux) are impressively higher. In addition, based on the fact that cell growth is enhanced with rising temperatures in the range of 14–35 °C that represents the viable range for microalgae [4], it is logical to expect even higher yields if we culture *Nephroselmis* using higher temperatures (e.g., 25–30 °C) than 20–21.5 °C used in the present study. Much future research is needed for the culture of *Nephroselmis* concerning the effect of many factors on its growth characteristics for establishing a reliable and trustworthy culture protocol.

*4.3. Tetraselmis sp. (var. red pappas)*

This strain of the species *Tetraselmis* sp. was given the arbitrary name "var. red pappas" due to its original location in the lagoon pappas in W. Greece. It is a chlorophyte that differs from other species of genus *Tetraselmis* with its peculiar and astonishing characteristic of imparting a dark-red coloration to its culture medium (Figure 10) in conditions of illumination of 8000 lux when it reaches the stationary phase, as observed in our culture. This unique phenomenon has never been reported in the literature and we cannot provide a plausible explanation other than it may be due to the accumulation of extracellular substances excreted by the senescent cells. It definitely cannot be attributed to carotenoids as the absorbance spectrum of the supernatant of centrifuged culture samples devoid of cells did not present any peaks corresponding to any kind of pigments, being almost absolutely zero-valued and flat along the range of 350–750 nm. So, we assume that natural excretion of organic matter by its intact or lysed cells may be the cause for this phenomenon as such excretions are known to occur in microalgae in various intensities depending on the species, conditions, and phase of the culture [31–36]. As the red coloration only occurs late in the stationary phase, it is probably due to high molecular substances (carbohydrates or humic species from decomposition of cells) [35,37,38] rather than low molecular ones (peptides or small proteins) that are known to be excreted during the exponential phase of the culture [31,37,38]. In the literature, there are various studies on the culture of several species of the genus *Tetraselmis* (*Tetraselmis* sp., *T. suecica*, *T. chui*) but they vary

substantially in the conditions used and the output data. What seems, from all of them, to be in accordance with our data is that *Tetraselmis* grows best at salinities of 35 ppt or lower and that high illumination results in being more productive [39–46]. This was confirmed in our study as the maximum density of *Tetraselmis* reached the level of ~$10 \times 10^6$ cells/mL in the salinities of 20 and 40 ppt and high illumination (8000 lux), far higher than that corresponding to 60 ppt (~$6 \times 10^6$ cells/mL) and even higher from their counterparts of low (2000 lux) illumination (~$4.1 \times 10^6$, $3.2 \times 10^6$, and $2.1 \times 10^6$ cells/mL at salinities of 20, 40 and 60 ppt respectively). Based on our experience of many years in culturing *Tetraselmis suecica* and now the present *Tetraselmis* sp. (var. red pappas), we have never experienced densities over $10 \times 10^6$ cells/mL. In this respect, the value of $35 \times 10^6$ cells/mL in [44] using low-intensity light of 1588 lux is, in our opinion, questionable. We also consider the notation of [42] that their culture of *T. suecica* entered the stationary phase on the fourth day and presented a growth rate of 0.8 yielding biomass dry weight of 0.57 g/L as questionable, because in our cultures, *Tetraselmis* kept its exponential phase at salinities of 20 and 40 ppt for 17 days and entered the stationary phase on day 14 only at the salinity of 60 ppt. Our higher yield in biomass of about 2.0 g d.w./L supports the overall published merits of the genus *Tetraselmis* as an ideal species for aquaculture feed [47], a candidate for biodiesel production [48], and easy to manipulate, due mainly to its high tolerance to extreme salinities [49,50].

### 4.4. Asteromonas gracilis

This extremely halotolerant (in respect to salinities higher than seawater) chlorophyte [15,16,51], which has been proven as suitable live food for rotifers in marine fish hatcheries [18,19] and a candidate for biofuel production [52], has drawn little attention for mass culture. To the best of our knowledge, our study is the first one focused on the basics of its growth in batch culture in order to be considered as a starting point for future more elaborate studies. Compared to other cultured microalgae, it is the biggest in cell size (18–22 μm) and because of this, its maximum density is around $6.5 \times 10^5$ cells/mL (Figure 15, right), which was attained at the highest (100 ppt) among the salinities tested and in high illumination (8000 lux). The salinity of 100 ppt in high light also presented the highest yield of 0.7 g/L as compared to 40 and 60 ppt (both ~0.5 g/L). All these values are higher than the only one reported in the literature [52] of ~0.4 g/L in the culture of which they conducted the experiment in higher temperature (25 °C, ours were 20–21.5 °C) but with weaker illumination (2500 lux, ours was 8000 lux for the abovementioned values) and recorded the end of exponential (log) phase on day 9, while in our culture, even on day 17, the exponential (log) phase had not yet ended. We feel that our results, apart from corroborating (at least in part) the findings of [52], must be considered as a more realistic approach due to the meticulous planning of our experimentation that had the sole purpose to study the parameters of growth of *A. gracilis*. We can certainly conclude that this species grows best at salinities over 60 ppt, and apart from its highest cell density and yield attained, we provide an indirect proof of it in having the shortest lag phase in 100 ppt (3 days) as compared to 5 and 8 days recorded in 60 and 40 ppt respectively. From all the above however, the opinion that *A. gracilis* cannot be cultured in salinities lower than 100 ppt must be discarded because as it is shown in Figure 15 (left and middle), it also grows well in both 40 and 60 ppt.

### 4.5. Dunaliella sp.

Species of the genus *Dunaliella* have been the subject of numerous studies over the years as this chlorophyte is notorious for its high salt tolerance by producing excessive amounts of glycerol and carotenoids [53,54], and aquaculturists aiming to the production of value-added products found it very useful long ago [55,56]. Being so, it comes of no surprise to find in the literature highly variable data concerning its growth performance in terms of density of culture (cells/mL), growth rate, and biomass yield as a consequence of the various volumes and conditions used in each particular experimentation. In the

present study our batch culture was an experiment using medium volumes (2L), easily attainable average temperature (20–21.5 °C) and light intensities (2000 and 8000 lux), simple aeration with no extra $CO_2$, a well-balanced nutrient mixture (Walne's), and three salinities covering the range from brackish to almost double-strength seawater. We found that our local strain of *Dunaliella* grew well in all salinities (20, 40, and 60 ppt) reaching more than $25 \times 10^6$ cells/mL after 3 weeks in light conditions of 8000 lux, and to this point, concerning the effect of light in particular, we are in accordance with most other studies [57–61] but in contrast to [62] in which growth decreased with increasing light intensity. Concerning the salinity, we demonstrated that this local strain has an impressive ability to osmoregulate and adapt to both low and high salinity, growing with the same speed in the range of 20–60 ppt. This clear-cut conclusion, supported by our results in high light (8000 lux) of final densities ($>25 \times 10^6$ cells/mL), SGR (0.350–0.405 doubl./day), and yield (1.25–1.58 g d.w./L), ranks our tested strain of *Dunaliella* among the most productive of its counterparts in the literature.

## 5. Conclusions

In all five species cultured, two things were common to all and deserve special attention. First, that growth was far better in high light (8000 lux) as compared to low light (2000 lux), and second, the elevated values of pH in high light throughout the exponential phase and then its decrease as the culture advanced in maturation. The intensity of photosynthesis is enhanced as light intensity increases and this was reflected in elevated pH values due to intense proton absorbance by the active cells [63]. The five species examined exhibited different responses to the salinities used, whereby *Amphidinium* clearly performs best in 20 ppt, far better than 40 ppt, and even more so than 50 ppt. *Nephroselmis* grows almost the same in 20 and 40 ppt and less well in 60 ppt, although the SGR for the first 10 days was almost the same for all salinities. Almost the same performance as *Nephroselmis* was shown by *Tetraselmis*. *Asteromonas* performs best in 100 ppt although it can grow quite well in both 40 and 60 ppt. *Dunaliella* grows equally well in all salinities (20, 40, and 60 ppt). Considering productivity as the maximum biomass yield at the end of the culture period, first rank is occupied by *Nephroselmis* with ~3.0 g d.w./L, followed by *Tetraselmis* (2.0 g/L), *Dunaliella* (1.58 g/L), *Amphidinium* (1.19 g/L), and *Asteromonas* (0.7 g/L). All these values resulted from batch cultures in a moderate temperature of 20–21.5 °C, and we speculate that at higher temperatures, yields could be even higher. For all five species, the literature has accumulated much research for various species of *Dunaliella* and *Tetraselmis*, very few for *Nephroselmis* and *Amphidinium*, and almost none for *Asteromonas*. However, in order to extract useful data from the existing bibliography for comparison with the data of the present work, the situation is very puzzling, as data on growth, SGR, and yield greatly varied, and often contradicted each other. This obviously results from the different conditions selected for their experimentation, the methodology, and time selected for the measurements. It is advised here for future works that the calculation of SGR should be made in a clearly indicated time period (i.e., from day "a" until day "b" of the culture), preferably from the day of the end of the lag (adaptation) phase until a day close to the middle of the log (exponential) phase.

**Supplementary Materials:** The following are available online at: https://youtu.be/3Vew3G9IRUE, Video S1: *Tetraselmis* sp. (var. red pappas). https://youtu.be/R8ue4H6zuYQ, Video S2: *Amphidinium carterae*. https://youtu.be/giZ430t15Sc, Video S3: *Nephroselmis* sp. https://youtu.be/_t8HNZ457XQ, Video S4: *Asteromonas gracilis*. https://youtu.be/a7X_0walwRQ, Video S5: *Dunaliella salina*.

**Author Contributions:** G.N.H.: Conceptualization, methodology, formal analysis, data curation, writing—original draft preparation, writing—review and editing, supervision, project administration, and funding acquisition. D.A.: Methodology, experimentation, and data curation. All authors have read and agreed to the published version of the manuscript.

**Funding:** This research was financially supported by the research program "ALGAVISION: Isolation and culture of local phytoplankton species aiming to mass production of antibacterial substances, fatty acids, pigments and antioxidants" (MIS 5048496), funded by the General Secretariat of Research and Technology of the Greek Government.

**Acknowledgments:** The authors thank the technical staff of the laboratory Athina Samara for her help in experimentation.

**Conflicts of Interest:** The authors declare no conflict of interest. The funders had no role in the design of the study; in the collection, analyses or interpretation of data; in the writing of the manuscript, or in the decision to publish the results.

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
