# Peer review of "The Effect of Various Salinities and Light Intensities on the Growth Performance of Five Locally Isolated Microalgae [Amphidinium carterae, Nephroselmis sp., Tetraselmis sp. (var. red pappas), Asteromonas gracilis and Dunaliella sp.] in Laboratory Batch Cultures"

_jmse, doi:10.3390/jmse9111275_

Round 1
Reviewer 1 Report
The current study discusses an important point in the field of microalgae culture, while significant major corrections are necessary before the publication can be accepted. The introduction is inadequate and didn’t write well. Many paragraphs were written without references; moreover, the authors should include some previous studies regarding the effect of different salinities and light on the species isolated in the current study. In the methodology Section, many data are missing like the culture time for all species which will be good if presented in Table, together with the experimental culture conditions for all species. In regarding to considered current study is the quantities study only, without any determination of the quality of isolated species, the maximum specific growth rate (μ) and doubling time (tg) cannot be considered to evaluate the quantities, at least authors should determine the biomass productivity (mg L-1 day-1) for each species, which they can find it here (https://doi.org/10.3390/su13041769). Moreover, It will be better if authors investigate the biochemical composition of isolated species to find out their importance and nutritional value. Another important point, authors must divide the M&M section into subtitles (2.1. Isolation and purification; 2.2. Experimental conditions; 2.3. Tested Parameters; and 2.4. Statistical Analysis), as well as, must include the references of the mentioned equations. The Tables from 1 to 5, must be rewritten. It should be only three rows: (1) the header, (2) marge SGR and SE together in one row (SGR±SE), and marge tg and SE together in one row (tg ±SE), while The rows of the period (days) and n must be moved as the footer of the tables, together with the title from “. The different superscripts indicate a statistically significant difference at the 0.05 ….. of the corresponding letter.). On all presented data in Figs, the author should determine the late exponential phase (LEP) due to its significant value in the growth curves. Discussions are informative and presented well, while authors should number the subtitles for all species. Please, find the corrections in the attached pdf.

Author Response
Dear reviewer thank you for your time and your constructive remarks on our document. I can feel that you have expertise on the subject so it will be great pleasure for us if I can satisfy your remarks to the point where it is possible and finally have your approvement.
I read carefully your report and tried to address every point. First of all I want to inform you that the whole experimentation was a time consuming (more than 6 months) painstaking path that followed another painstaking procedure of more than a year collection of lagoonal samples and continuous subcultures and renovations of cultures in order to find microalgae that could be effectively cultured. Dozens of cultures of other various species had been discarded in the time path as not suitable for mass culture till we selected the ones we present in our paper.
Because of the above, the primary task of us was to find the basic response of them in terms of growth and yield in biomass under basic conditions of salinity and light. As I mention in the introduction (Lines 49-59) if a cultured species cannot grow well there is little importance to explore its biochemical profile or investigate its response to various parameters. Now that we have completed the first task of growth we'll enter in another experimentation concerning the effect of e.g. color of illumination, deprivation of nitrogen etc on its biochemical profile for each one species at the most effective light intensity and salinity found in the present paper. In this respect I wish for your understanding for not providing here results of biochem. profiles, Also as the experimentation has ended it is impossible now to have data on biomass productivity (mg L-1 day-1) like the ones you suggested with the really valuable paper you cited which we have studied it with pleasure.
Concerning the introduction I responded to the maximum point possible to your remarks inserting additional references in Lines 41, 48. Concerning citations in Lines 60, 65 and 74 you suggested I feel sorry to inform you that I cannot find a suitable one for each case because: In the case of Line 60 where I write ".... considered as a default parameter" it is purely my opinion and I considered it as a personal contribution to experiments of this kind. In the case of Line 65 concerning the beneficial effect of lack of heat emmitted by LED light I did not find any literature exactly on this point so I write on this as "personal observation". In Line 74 where I write " and beyond that, the influence of these on the biochemical composition" it is again my personal approach in expressing my personal opinion so I cannot cite something in this paragraph.
Concerning the M&M I think I responded fully to your suggestions. I inserted 2 citations in Lines 106 and 119 you indicated and I have reconstructed the chapter by having divided it into subtitles of Isolation and purification (a new one), Experimental conditions, and Tested parameters and statistical analysis (I think it is better to unite them in a single subtittle).
I reconstructed all the Tables the very useful way you suggested and thank you very much for it.
The culture time for each culture is clearly written in the text and moreover it is clearly shown in the Figures. Also concerning the late exponential phase you mention it is clearly indicated in the graphs and mentioned many times in the text at relevant points of the calculation of SGR. I really thought a lot about your remark and I was greatly puzzled what addition could be done to indicate that on the graphs. Finally I think it is better not to load the already densely packed with information graphs with new indicative material and I hope to have your understanding on this issue also.
Concerning your remark in the introduction section as to include previous studies regarding the effect of different salinities and light on the species isolated in the current study I can inform you that there are none about our species (with the exception of Dunaliella which I deserved them for the Discussion section), so our study comes to fill this gap.
We are engaged in batch culturing microalgae for more than 25 years and at present we focus on local species. Some of them may be good candidates for mass culturing. We are procceeding step by step. Please consider our novel local species as worth studying and give us your approval to publish our first results.
In the attached file there are all the above corrections. Thank you very much for your expertise conveyed to me.
Reviewer 2 Report
In my opinion authors should improve a Introduction. It is not clear why that species were investigated. Authors said about valuable products but only biomass was analysed. Authors compared only two level of light. As it was mentioned in article level of light is very important for economical aspect of algae cultivation. So why ony two steps were analised. Maybe 10000lux will be muche better or won't be diferences between 6000 lux and 8000lux. Of course in the litheratura it is possible find a lot of information about light influece. Authors look for answers for way of cultivation a new algaes speciess but they should explein why this new speciess are interesting.
The main question is why authors used that same cultivation cnditions for different speciess. Maybe they should optamlising conditions for specific species. In the discusion is showned that many of information about solnity and lights is vell known. Please point what is realy new in your manuscript.
Author Response
Dear reviewer thank you for your time and your constructive remarks on our document. It will be great pleasure for me if I can satisfy your remarks to the point where it is possible and finally have your approvement.
I read carefully your report and tried to address every point. First of all I want to inform you that the whole experimentation was a time consuming (more than 6 months) painstaking path that followed another painstaking procedure of more than a year collection of lagoonal samples and continuous subcultures and renovations of cultures in order to find microalgae that could be effectively cultured. Dozens of cultures of other various species had been discarded in the time path as failed in establishing a stable population and judged as not suitable for mass culture till we selected the ones we present in our paper.
Because of the above, the primary task of us was to find the basic response of them in terms of growth and yield in biomass under basic conditions of salinity and light. As I mention in the introduction (Lines 49-59) if a cultured species cannot grow well there is little importance to explore its biochemical profile or investigate its response to various parameters. Now that we have completed the first task of growth we'll enter in another experimentation concerning the effect of e.g. more light intensities, color of illumination, deprivation of nitrogen etc on its biochemical profile for each one species at the most effective light intensity and salinity found in the present paper. In this respect I wish for your understanding for not providing here results of biochem. profiles.
Of course you are absolutely right stating that light is of paramount importance. The range of light intensities to be investigated can be very wide but it was impossible for us to check a lot of them, so we carefully selected two completely different light regimes that of a low 2000 lux and that of a rather high 8000 lux to check (to begin with them) the response of the 5 species. The results were clear and lead to substantial conclusions.
There is very limited (or none) information in the literature (the only exception for the Genus Dunaliella) about the culture of the species we examined. In reality and frankly speaking for Amphidinium carterae, Nephroselmis, Tetraselmis (this peculiar species of ours) and Asteromonas there are no information like that we are presenting in our paper. We wish to pave the way for further studies for these strains (they are local strains and we plan to deepen further in them).
We used the same conditions we created for all species because they are a good approximation to ordinary situations. They can be considered as default to begin with. We cannot know from the beginning what special condition an algae prefers for maximizing production. We can not at the same time try a lot of conditions. This will be done in future trials. We had 5 species and they loaded already the paper.
Of course in the discussion there is a welth of information about light and salinity but is so fragmented and scattered that we incorporated them to extract what could be really connected to our findings and explain (less or more) them. Everything is new in our paper. First of all there are 5 local species that have not been batch cultured in a clear cut way we did. Even for the most studied Dunaliella species we found and elucidated clearly that our local species exhibited an impressive growth potential. For the other 4 species it goes without saying that the information we present is novel. So I think that all these issues were addressed adequately in the Discussion.
Finally I can state here that these 5 species are interesting because: 1} they are local strains from lagoons of W. Greece. 2} They ensued as culturable from dozen other species (from our samples) that were tested for culture and failed. 3) They proved to grow well in our easily attainable in every laboratory basic (lets say default) conditions and beyond that we know now the best salinity and light intensity yielding maximum biomass (of course more experimentation is pending). It will be great pleasure for us to supply you (if you wish) pure culture samples of the 5 species in order to share with other colleagues our material because I believe in the open borders of science.
I dont know what else could I write here in response to your valuable remarks. We invested much-much time and effort for this initial study that ensued from more than a year (additionally) screening for microalgae the local lagoons. I hope your expertise to give it approval for publication so as to continue further.
I attach here the revised document with a new section in the Materials and Methods about isolation and purification, some more citations in the introduction and some reconstructions to be more practical in the Tables.
Round 2
Reviewer 1 Report
all necessary corrections have been made by authors, while three needed Refs are still required in the introduction section. However, I accept the manuscript in the current form, and I ask the authors to add these refs.
These refs are:
1. P2L59
2. P2L65 ( There are many papers and reports that concluded the same of authors' personal observation, so, it would be good if authors add a reference)
3. P2L73
Reviewer 2 Report
Manuscript can be published